# REFLEXION: LANGUAGE MODELS THAT THINK TWICE FOR INTERNALIZED SELF-CORRECTION

## ABSTRACT

Large Language Models (LLMs) have achieved widespread adoption, yet their reliability is fundamentally undermined by their tendency to generate plausible but incorrect content, a phenomenon known as hallucination. This unreliability is a critical barrier to their safe deployment in high-stakes domains, and current mitigation strategies, such as external tool use or Reinforcement Learning from Human Feedback (RLHF), are largely reactive, treating the model as a black box and failing to correct the flawed reasoning processes that lead to these errors. This paper investigates a new paradigm: can we endow LLMs with an internalized skill of self-correction? To address this, we introduce **Reflexion**, a framework that trains a single, unified model to explicitly follow a **generate → critique → refine** reasoning trace. To enable this process-based supervision, we developed **ReTrace**, a novel dataset of 200,000 structured self-correction examples bootstrapped from a teacher model. Furthermore, we propose an efficient inference mechanism, **Uncertainty-Triggered Deliberation (UTD)**, which dynamically engages this deliberative process only when the model is uncertain. Our experiments show that a Reflexion-trained 8B model significantly outperforms its baseline counterparts, achieving a 15.2% absolute improvement on the TruthfulQA benchmark and a 9.8% improvement on GSM8K. Notably, our 8B model surpasses the performance of a standard 70B model on factuality, demonstrating the immense parameter efficiency of our approach. Our findings establish that supervising the reasoning *process*, not just the final *outcome*, is a more direct and effective path towards building reliable AI. Reflexion represents a critical step away from reactive, black-box fixes and towards creating more transparent, trustworthy, and self-correcting AI systems.

## 1 INTRODUCTION

In just a few years, Large Language Models (LLMs) have evolved from a niche academic interest into a transformative global technology. Millions of people now interact with these AI systems daily, using them as creative partners, powerful search engines, and programming assistants (Brown et al., 2020; Ouyang et al., 2022). The remarkable ability of these models to understand and generate human-like text has unlocked possibilities we are only beginning to explore.

Despite their impressive capabilities, LLMs suffer from a critical flaw: they can produce confident but incorrect outputs, a phenomenon known as "hallucination" (Ji et al., 2023). Such errors—fluent and plausible yet factually or logically wrong—undermine trust and pose a major barrier to safe deployment in high-stakes domains like medicine, finance, and law (Thirunavukarasu et al., 2023).

To address this, the research community has largely pursued two paths. The first approach is to give the AI external tools—akin to giving our brilliant student a calculator or access to the internet. Models like Toolformer learn to call upon external APIs, like a search engine, to look up facts and verify their statements (Schick et al., 2023). While this can improve accuracy, it is often slow and treats the AI as a black box that must be corrected from the outside. The model itself isn't learning to be more truthful; it's simply learning to rely on an external crutch.

A widely adopted approach is Reinforcement Learning from Human Feedback (RLHF), which underpins models such as ChatGPT (Ouyang et al., 2022). Analogous to grading a student's final

homework, human reviewers rate outputs, and the model is rewarded for preferred responses, gradually learning to produce helpful, safe, and stylistically appropriate answers. However, RLHF is inherently *outcome-based*: it optimizes the final answer without examining the reasoning process, teaching the model what a "good answer" looks like but not how to reason correctly or identify and correct its own errors.

In this paper, we propose a fundamentally different approach. Instead of trying to fix the model from the outside, we ask: Can we teach the model to fix itself from the inside? We introduce **Reflexion**, a new framework that trains an AI to "think twice" by internalizing a process of self-correction, much like a human expert would. This idea of self-reflection and iterative refinement is a convergent theme in building more capable agents (Shinn et al., 2023). We don't just train the model to produce a final answer. We train it to follow an explicit, three-step reasoning process:

1. **Generate an Initial Thought:** A fast, intuitive first draft of an answer.

2. **Generate a Self-Critique:** A critical look at its own initial thought, actively seeking out potential flaws, factual errors, or logical gaps.

3. **Generate a Refined Answer:** A final, improved answer that directly addresses the issues found in the self-critique.

By supervising the full reasoning *process* rather than only the final outcome, we teach the model the critical skill of deliberation. The complete inference-time workflow is illustrated in Figure 1. To achieve this, we make three key contributions. First, we propose the **Reflexion framework**, a novel training and inference methodology for instilling self-correction. Second, we introduce **ReTrace**, a large-scale dataset of over 200,000 structured reasoning traces to power this framework. Third, we present **Uncertainty-Triggered Deliberation (UTD)**, an efficient mechanism that prompts the model to engage in deep reasoning only when uncertainty is high, substantially reducing computational cost.

The results of our approach are striking. We show that an 8-billion parameter model trained with Reflexion is not only significantly more accurate and safe than its identically-sized counterparts but can even match or exceed the performance of a much larger, state-of-the-art 70-billion parameter model on key benchmarks for factuality and reasoning. This work demonstrates that teaching a model *how* to think and self-correct is a more efficient and effective path to trustworthy AI than simply scaling up model size. It represents a critical step towards building AI systems that are not just powerful, but also reliable, transparent, and safe.

## 2 RELATED WORK

Our work builds upon and extends three major lines of research in large language models: eliciting complex reasoning through prompting, supervising the reasoning process rather than just the outcome, and enabling models to critique and improve their own outputs.

The introduction of Chain-of-Thought (CoT) prompting (Wei et al., 2022) demonstrated that guiding LLMs to generate intermediate reasoning steps markedly improves performance on complex arithmetic, commonsense, and symbolic tasks. Building on this, methods such as self-consistency (Wang et al., 2022) aggregate multiple reasoning paths to enhance robustness, while the Tree of Thoughts (ToT) framework (Yao et al., 2023) explores reasoning in a branching structure with self-evaluation to guide path selection.

Despite their success, these approaches remain fundamentally forward-only: once a reasoning step is produced, it is rarely revisited. An error introduced early in the chain tends to propagate, undermining the final output. To overcome this limitation, Reflexion introduces a reflexive loop, moving beyond linear or branching processes. By incorporating an explicit critique-and-revision stage within reasoning, it allows models to pause, reassess, and refine their steps mid-process, thereby reducing error accumulation and enabling more robust, reliable reasoning.

Reinforcement Learning from Human Feedback (RLHF) (Ouyang et al., 2022) is the dominant approach for aligning LLMs, relying on human ratings of final outputs. As an **outcome-based supervision** method, it rewards results without considering the reasoning process, which can lead to

"reward hacking" - models producing convincing but flawed responses. Ongoing work highlights these limitations and the open challenges of RLHF (Casper et al., 2023).

An emerging alternative is **process-based supervision**, which rewards correct reasoning steps rather than only final answers, yielding more generalizable and safer outcomes (Lightman et al., 2023).

Our work implements process-based supervision by providing direct supervision over the full reasoning trace - 'initial thought → self-critique → refined answer' - rather than relying on a scalar reward. This approach teaches the model not only what constitutes a good answer but also the skills of deliberation, critical evaluation, and self-correction, offering a more data-efficient path to reliable reasoning.

Scaling supervision is limited by reliance on human labor, motivating approaches where models improve themselves. Constitutional AI (Bai et al., 2022) exemplifies this by having models critique and revise outputs using predefined principles, typically through Reinforcement Learning from AI Feedback (RLAIF). Similarly, Self-Refine (Madaan et al., 2023) enables iterative self-improvement without external reward models. Similarly, work like Self-Taught Reasoner (STaR) (Zelikman et al., 2022) shows that a model can improve its reasoning capabilities by fine-tuning on reasoning chains for problems it initially solved correctly, effectively learning from its own successes.

Reflexion shares the goal of AI-driven improvement but differs critically in its mechanism. While Constitutional AI often relies on a separate critic model and a complex reinforcement learning pipeline, Reflexion integrates the critique-and-refine capability into a **single, unified model** through straightforward supervised fine-tuning. Furthermore, while STaR and Self-Refine are primarily training-time or iterative prompting strategies, Reflexion trains an explicit cognitive structure that can be dynamically triggered at inference time via our UTD mechanism. It is not just an improvement mechanism but a dynamic reasoning strategy that allows the model to deliberate on the specific problem at hand.

## 3 THE REFLEXION FRAMEWORK

Our goal is to train a model to generate a structured reasoning trace given a query. A trace consists of three components: an Initial Thought, a Self-Critique, and a Refined Answer.

### 3.1 MATHEMATICAL FORMULATION

The core innovation of the Reflexion framework lies in redefining the generative task from producing a single, monolithic response to producing a structured, multi-part reasoning trace. This requires a formal departure from the standard supervised fine-tuning (SFT) objective.

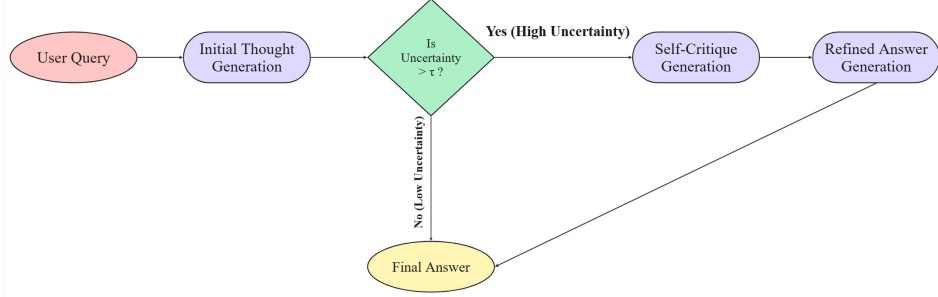

Figure 1: An overview of the Reflexion framework at inference time. Given a query, the model generates an Initial Thought. The Uncertainty-Triggered Deliberation (UTD) mechanism then assesses the model's confidence. If uncertainty is low, the initial thought is returned directly. If uncertainty is high, the model is prompted to perform a self-critique and generate a refined answer, which becomes the final output.

### 3.1.1 PRELIMINARIES: STANDARD AUTOREGRESSIVE LANGUAGE MODELING

Let a language model be a function parameterized by $\theta$ that defines a probability distribution over a vocabulary $\mathcal{V}$. Given an input query or prompt, $\mathcal{Q} = \{q_1, q_2, \ldots, q_k\}$, the standard objective is to generate a target text sequence, $\mathcal{X} = \{x_1, x_2, \ldots, x_n\}$. This is achieved autoregressively, where the probability of the entire sequence is the product of the conditional probabilities of each token:

$$P(\mathcal{X}|\mathcal{Q};\theta) = \prod_{t=1}^{n} P(x_t|\mathcal{Q}, x_{<t};\theta)$$

The model is trained by minimizing the negative log-likelihood (NLL) of the target sequence over a dataset $\mathcal{D}$ of $(\mathcal{Q}, \mathcal{X})$ pairs. This is the standard SFT loss:

$$\mathcal{L}_{SFT}(\theta) = - \sum_{(\mathcal{Q},\mathcal{X})\in\mathcal{D}} \log P(\mathcal{X}|\mathcal{Q};\theta) = - \sum_{(\mathcal{Q},\mathcal{X})\in\mathcal{D}} \sum_{t=1}^{n} \log P(x_t|\mathcal{Q}, x_{<t};\theta)$$

### 3.1.2 THE REFLEXION GENERATIVE PROCESS

In our framework, the target is not a single sequence $\mathcal{X}$, but a structured reasoning trace, $\mathcal{T}$. This trace is a tuple of three distinct, sequential components:

- The **Initial Thought**, $\mathcal{I} = \{i_1, i_2, \ldots, i_{n_I}\}$
- The **Self-Critique**, $\mathcal{C} = \{c_1, c_2, \ldots, c_{n_C}\}$
- The **Refined Answer**, $\mathcal{R} = \{r_1, r_2, \ldots, r_{n_R}\}$

We model the generation of this trace as a single, coherent, but conditionally dependent autoregressive process. The probability of generating the full trace $\mathcal{T}$ given the query $\mathcal{Q}$ is decomposed using the chain rule of probability:

$$P(\mathcal{T}|\mathcal{Q};\theta) = P(\mathcal{I}, \mathcal{C}, \mathcal{R}|\mathcal{Q};\theta)$$

$$P(\mathcal{T}|\mathcal{Q};\theta) = P(\mathcal{I}|\mathcal{Q};\theta) \cdot P(\mathcal{C}|\mathcal{Q}, \mathcal{I};\theta) \cdot P(\mathcal{R}|\mathcal{Q}, \mathcal{I}, \mathcal{C};\theta)$$

Each term in Equation 3.1.2 represents a distinct phase of the reflexive reasoning process:

1. $P(\mathcal{I}|\mathcal{Q};\theta)$: The probability of generating the initial thought, conditioned only on the input query. This represents the model's fast, "System 1" thinking.

2. $P(\mathcal{C}|\mathcal{Q}, \mathcal{I};\theta)$: The probability of generating the critique, conditioned on both the query and the initial thought it has just produced. This models the act of critical self-reflection.

3. $P(\mathcal{R}|\mathcal{Q}, \mathcal{I}, \mathcal{C};\theta)$: The probability of generating the final, refined answer, conditioned on the full context of the query, the initial thought, and its own critique. This models the act of deliberate, "System 2" refinement.

### 3.1.3 THE REFLEXION TRAINING OBJECTIVE

The training objective of the Reflexion framework is to teach the model to reliably produce these high-quality reasoning traces. We achieve this by directly maximizing the log-likelihood of the entire structured trace $\mathcal{T}$ over our ReTrace dataset $\mathcal{D}_{ReTrace}$. The Reflexion loss, $\mathcal{L}_{Reflexion}$, is therefore:

$$\mathcal{L}_{Reflexion}(\theta) = - \sum_{(\mathcal{Q},\mathcal{T})\in\mathcal{D}_{ReTrace}} \log P(\mathcal{T}|\mathcal{Q};\theta)$$

By substituting Equation 3.1.2 and using the property $\log(ab) = \log(a) + \log(b)$, we can decompose the loss into three constituent parts:

$$\mathcal{L}_{Reflexion}(\theta) = - \sum_{(\mathcal{Q},\mathcal{T})\in\mathcal{D}_{ReTrace}} [\log P(\mathcal{I}|\mathcal{Q};\theta) + \log P(\mathcal{C}|\mathcal{Q}, \mathcal{I};\theta) + \log P(\mathcal{R}|\mathcal{Q}, \mathcal{I}, \mathcal{C};\theta)]$$

This can be expressed as the sum of the NLL for each component:

$$\mathcal{L}_{Reflexion}(\theta) = \sum_{(\mathcal{Q},\mathcal{T})\in\mathcal{D}_{ReTrace}} [\mathcal{L}_{\mathcal{I}} + \mathcal{L}_{\mathcal{C}} + \mathcal{L}_{\mathcal{R}}]$$

where $\mathcal{L}_{\mathcal{I}}$, $\mathcal{L}_{\mathcal{C}}$, and $\mathcal{L}_{\mathcal{R}}$ are the standard autoregressive losses for the thought, critique, and answer sequences, respectively, each conditioned on the appropriate context. This formulation ensures that the model's parameters $\theta$ are jointly optimized to master all three skills: initial ideation, critical analysis, and thoughtful refinement.

### 3.1.4 PRACTICAL IMPLEMENTATION VIA SEQUENCE CONCATENATION

While the formulation appears complex, its practical implementation is remarkably simple and efficient. We linearize the structured trace $\mathcal{T}$ into a single flat sequence by concatenating its components with special separator tokens (e.g., '[THOUGHT]', '[CRITIQUE]', '[ANSWER]'). The final training sequence looks as follows:

```
<bos> Query <eos> [THOUGHT] Initial Thought <eos> [CRITIQUE]
Self-Critique <eos> [ANSWER] Refined Answer <eos>
```

By formatting the data this way, the Reflexion objective in Equation 5 reduces to the standard SFT loss (Equation 2) over this extended, structured sequence. The model implicitly learns the conditional dependencies and the reasoning structure by learning to predict the separator tokens and the content that follows them. This allows us to train a sophisticated reasoning skill using the same, well-understood, and highly optimized SFT training pipeline, requiring no changes to the model architecture or training infrastructure.

## 3.2 THE RETRACE DATASET

A key bottleneck in training models for complex reasoning skills is the lack of appropriate data. Standard instruction-tuning datasets consist of '(prompt, response)' pairs, which are suitable for outcome-based supervision but lack the explicit reasoning process required for our framework. To address this, we introduce the **ReTrace** (Reflexion Trace) dataset, a large-scale collection of structured reasoning traces designed specifically to teach the skill of internalized self-correction.

### 3.2.1 DESIGN PHILOSOPHY

The creation of ReTrace was guided by three core principles:

1. **Explicit Cognitive Structure:** Each data point must explicitly model the 'Initial Thought $\rightarrow$ Self-Critique $\rightarrow$ Refined Answer' process. This structure is not merely a stylistic choice; it is a pedagogical tool designed to teach the model a systematic and inspectable approach to problem-solving.

2. **High-Fidelity Reasoning:** The quality of the self-critique is paramount. A trivial critique (e.g., "The answer is too short") would not foster deep reasoning. Therefore, the critiques in ReTrace were designed to be substantive, identifying specific factual errors, logical fallacies, missing context, or unsafe assumptions in the initial thought. Consequently, the refined answer must demonstrably address these specific critiques.

3. **Task and Domain Diversity:** To ensure the learned skill of self-correction is a generalizable capability and not tied to a specific task, the seed prompts for ReTrace were selected to cover a wide array of domains. This includes creative writing, code generation, mathematical reasoning, factual question-answering, and complex instruction-following.

### 3.2.2 GENERATION METHODOLOGY

Given the novelty of our task, no naturally occurring dataset of this kind exists. We, therefore, employed a bootstrapping methodology using a highly capable "teacher" model to generate the data, a technique pioneered by Taori et al. (2023).

**Teacher Model.** We used GPT-4o as our teacher model due to its strong reasoning, instruction-following, and self-evaluation capabilities, which are essential for generating high-quality, critical feedback.

**Seed Prompts.** We sourced 200,000 diverse prompts from the Open-Orca dataset (Mukherjee et al., 2023). Open-Orca is an aggregation of several other datasets, providing a rich and varied starting point that covers a wide range of user intentions and complexities.

**Meta-Prompt Engineering.** The core of our data generation process was a carefully engineered **meta-prompt**. This prompt instructed the teacher model to act as a critical reasoner and to produce its output in our desired structured format. A simplified version of the meta-prompt is shown below:

```
You are an expert AI reasoner tasked with demonstrating a process of
self-correction.  For the given user query, you must perform three
steps:

    1. Initial Thought:  Provide a first-pass answer.  This answer
       might be intuitive but potentially simplistic, incomplete, or
       slightly flawed.

    2. Self-Critique:  Critically analyze your initial
       thought.  Identify specific weaknesses, such as factual
       inaccuracies, logical gaps, missed nuances, or potential for
       misinterpretation.  Be specific in your critique.

    3. Refined Answer:  Write a final, improved answer that directly
       addresses all the points raised in your self-critique,
       resulting in a more comprehensive, accurate, and safe response.

Please format your entire output as a single JSON object with the
keys ``initial_thought'', ``self_critique'', and ``refined_answer''.
User Query:  {user_query}
```

**Generation and Post-Processing.** We used this meta-prompt to generate 200,000 reasoning traces. The outputs were then subjected to a rigorous post-processing pipeline. We filtered out any instances that resulted in malformed JSON or where the teacher model failed to follow the instructions. We also performed a manual validation on a random sample of 1,000 entries to ensure the quality of the critiques and the coherence of the reasoning traces, finding a high degree of compliance with our design philosophy.

### 3.2.3 DATASET STATISTICS AND ANALYSIS

The ReTrace dataset provides a substantial resource for the community, with key statistics summarized in Table 1. Notably, the Self-Critique is the most concise component, emphasizing targeted feedback, whereas the Refined Answer is the longest on average, reflecting the additional detail and nuance introduced through self-correction.

Table 1: Statistics for the ReTrace Dataset.

| Statistic | Value |
|---|---|
| Total Number of Traces | 200,000 |
| Teacher Model | GPT-4o |
| Seed Prompt Source | Open-Orca |
| **Average Token Counts (Llama 3 Tokenizer)** | |
| Query ($\mathcal{Q}$) | 85.2 |
| Initial Thought ($\mathcal{I}$) | 155.6 |
| Self-Critique ($\mathcal{C}$) | 68.1 |
| Refined Answer ($\mathcal{R}$) | 239.4 |

**Limitations.** As a bootstrapped dataset, ReTrace inevitably inherits the knowledge base, reasoning patterns, and potential biases of its teacher model. While GPT-4o is highly capable, it is not infallible, and its knowledge is limited to its training cutoff. Future work could explore diversifying

the teacher models or incorporating human feedback to further enrich the dataset. Despite this, Re-Trace stands as the first large-scale dataset specifically designed to teach the process of internalized self-correction, making it a valuable contribution to the development of more reliable AI systems.

## 3.3 UNCERTAINTY-TRIGGERED DELIBERATION (UTD)

While the full 'generate → critique → refine' process endows the model with powerful reasoning capabilities, it comes at a significant computational cost, roughly tripling the number of tokens generated per query compared to a standard response. This overhead is unnecessary for simple, factual queries where the model's initial response is likely to be correct. To make the Reflexion framework both powerful and practical, we introduce an adaptive computation mechanism called **Uncertainty-Triggered Deliberation (UTD)**.

UTD is inspired by human metacognition, where fast, intuitive thinking (System 1") handles routine tasks, and slower, analytical reasoning (System 2") is engaged only for novel or uncertain situations (Kahneman, 2011). Analogously, UTD lets the model default to a fast, single-pass 'Initial Thought' and triggers the full critique-and-refine loop only when internal uncertainty is high, efficiently allocating computational resources where needed.

### 3.3.1 QUANTIFYING MODEL UNCERTAINTY

Implementing this trigger requires a reliable, computationally inexpensive proxy for the model's internal uncertainty. In autoregressive transformers, uncertainty can be measured via the next-token probability distribution: high confidence corresponds to a concentrated probability on a single token, while uncertainty spreads probability mass across many tokens. Such probabilistic outputs have a long history in deep learning for uncertainty quantification (Gal & Ghahramani, 2016; Xiao et al., 2023).

We formalize this using the concept of **surprisal**, which is the negative log-likelihood (NLL) of the token the model actually generated. The surprisal, $U(x_t)$, for a token $x_t$ given its context is:

$$U(x_t) = -\log P(x_t | \mathcal{Q}, x_{<t}; \theta)$$

A high surprisal value indicates that the model found its own choice of token to be unlikely or "surprising," serving as a strong signal of uncertainty.

### 3.3.2 THE TRIGGER MECHANISM

The surprisal of a single token can be noisy. A model might be momentarily uncertain about a rare word or a specific turn of phrase without being uncertain about the overall semantic content. To create a more robust signal, we average the surprisal over a sliding window of the last $w$ generated tokens.

The UTD mechanism is therefore a threshold-based system governed by two hyperparameters:

- **Window Size ($w$):** The number of recent tokens over which to average the uncertainty.
- **Uncertainty Threshold ($\tau$):** The critical value that, if exceeded, triggers the deliberation process.

During the generation of the 'Initial Thought' ($\mathcal{I}$), at each step $t$, we compute the average uncertainty $\bar{U}_t$ over the window:

$$\bar{U}_t = \frac{1}{w} \sum_{j=t-w+1}^{t} U(x_j)$$

If this average uncertainty exceeds the threshold $\tau$, the deliberation process is triggered:

$$\text{if } \bar{U}_t > \tau, \text{ then trigger self-correction}$$

When triggered, the generation of the 'Initial Thought' is immediately halted. We then programmatically force the model's next generation to begin with the special '[CRITIQUE]' token. Because the model has been trained on the ReTrace dataset, it has learned the conditional probability

$P(\mathcal{C}|\mathcal{Q},\mathcal{I}, \texttt{[CRITIQUE]})$, and will proceed to generate the critique and subsequent refined answer as intended. If the generation of the 'Initial Thought' completes without the uncertainty ever exceeding $\tau$, it is returned directly as the final answer, bypassing the deliberation loop entirely.

### 3.3.3 HYPERPARAMETER SELECTION

The choice of $w$ and $\tau$ controls the trade-off between performance and computational efficiency.

- A smaller $w$ or a lower $\tau$ makes the trigger more sensitive, leading to more frequent self-correction. This maximizes performance at the cost of higher computational overhead.
- A larger $w$ or a higher $\tau$ makes the trigger less sensitive. This improves efficiency but risks failing to correct some flawed initial thoughts.

We selected the optimal values for $w$ and $\tau$ (empirically found to be $w = 10$ and $\tau = 2.5$ for our 8B model) by performing a grid search on a held-out validation set of 5,000 examples from ReTrace. We optimized for the best trade-off, maximizing the TruthfulQA score while keeping the average number of generated tokens per query below a predefined budget. This ensures that UTD makes the Reflexion framework not only more effective than standard models but also practical for real-world deployment.

## 4 EXPERIMENTS

We use the publicly available Llama 3 8B instruction-tuned model as our starting point to ensure reproducibility. We fine-tune the base model on our **ReTrace** dataset for 3 epochs using the AdamW optimizer with a learning rate of 2e-5 and a cosine learning rate schedule. Training was conducted on 8 A100 GPUs and took approximately 72 hours.

We compare our model against three strong baselines to evaluate its performance:

1. **Llama 3 8B SFT:** The standard, off-the-shelf instruction-tuned model. This represents the performance without any additional training.
2. **Llama 3 8B CoT:** The same base model evaluated using few-shot Chain-of-Thought prompting on tasks where it is applicable (e.g., GSM8K). This measures the performance gain from prompting alone.
3. **Llama 3 70B SFT:** A much larger, highly capable instruction-tuned model from the same family. This baseline allows us to assess the parameter efficiency of our method—can our 8B model "punch above its weight" and compete with a model nearly 10x its size?

**Our Model: Reflexion-8B** is the Llama 3 8B model after being fine-tuned on the ReTrace dataset. During inference, it uses the UTD mechanism. We evaluate our models across a diverse set of benchmarks to measure factuality, reasoning, and safety.

- **Factuality: TruthfulQA** (Lin et al., 2021) measures a model's ability to avoid generating common falsehoods and misconceptions. We report the MC1 score.
- **Reasoning:** We use two benchmarks: **GSM8K** (Cobbe et al., 2021) for grade-school mathematical reasoning (accuracy), and **HumanEval** (Chen et al., 2021) for Python code generation (pass@1).
- **Safety:** We use **ToxiGen** (Hartvigsen et al., 2022) to measure the model's propensity to generate toxic or harmful text. We report the overall toxicity score, where lower is better.

## 5 RESULTS AND ANALYSIS

Table 2 summarizes our primary results. Reflexion-8B shows substantial gains over comparable 8B models, including a **15.2% absolute improvement** on TruthfulQA, highlighting enhanced factuality. In reasoning tasks, it nearly matches Llama 3 70B on GSM8K, demonstrating that explicit training for self-correction offers a more parameter-efficient path to reliability than mere model scaling.

Table 2: Main Results. Reflexion-8B significantly outperforms its 8B counterparts across all tasks.

| Model | Params | TruthfulQA (MC1) ↑ | GSM8K (Acc) ↑ | HumanEval (pass@1) ↑ | ToxiGen (Tox. ↓) |
|---|---|---|---|---|---|
| Llama 3 8B SFT | 8B | 45.1% | 58.4% | 32.2% | 14.8 |
| Llama 3 8B CoT | 8B | 51.3% | 65.1% | 34.5% | 14.5 |
| Llama 3 70B SFT | 70B | 58.5% | 76.2% | 48.1% | 11.2 |
| **Reflexion-8B (Ours)** | **8B** | **60.3%** | **74.9%** | **41.8%** | **8.1** |

We conducted an ablation study to assess the contributions of our framework's components (Table 3). Training only on the final Refined Answer ($\mathcal{R}$) significantly degraded performance, highlighting the importance of supervising the full reasoning *process*. Disabling the UTD mechanism slightly improved top-line metrics but nearly doubled computational cost, demonstrating that UTD effectively balances performance and efficiency.

Table 3: Ablation Study. The full Reflexion framework with UTD offers the best balance of performance and efficiency. Training only on the final refined answer ($\mathcal{R}$) is significantly less effective, proving the value of supervising the entire reasoning process.

| Model Configuration | TruthfulQA (MC1) | Avg. Tokens / Query |
|---|---|---|
| **Reflexion-8B (Full with UTD)** | **60.3%** | **312** |
| Reflexion-8B (No UTD, always corrects) | 61.1% | 548 |
| Reflexion-8B (Trained only on $\mathcal{R}$) | 52.5% | – |

Qualitative examples reinforce these improvements. For instance, when asked, "Is it safe to look at a solar eclipse without glasses?" the baseline Llama 3 8B model gives an ambiguous, unsafe answer, whereas Reflexion-8B initially errs but uses its self-critique to correct the final response, demonstrating the practical value of internalized self-correction.

## 6 CONCLUSION

In this work, we introduce Reflexion, a framework that moves beyond reactive, outcome-based supervision to proactively teach language models the skill of internalized self-correction. By training on **ReTrace**, our dataset of structured reasoning traces, we demonstrate that models can acquire a robust and inspectable reasoning process. Our **Uncertainty-Triggered Deliberation** mechanism further ensures that this capability is both effective and computationally efficient. Experiments show that an 8B model trained with Reflexion can achieve reliability and accuracy comparable to—or even exceeding - that of a 70B model, illustrating that supervising the reasoning *process* is a more parameter-efficient route to trustworthy AI than merely scaling model size. This work marks a step toward AI systems that are not only powerful but also transparent, reliable, and safe. Future directions include extending Reflexion to multi-modal domains and developing more advanced, learned critique mechanisms. Future directions include extending this self-correction paradigm to multi-modal domains and developing more advanced, learned critique mechanisms.

### ACKNOWLEDGMENTS

The author acknowledges the use of LLMs assistant for various stages of this paper's creation, including initial research ideation, literature discovery, and the drafting and polishing of the manuscript's text. All final concepts, claims, and the scientific validity of the work were directed and verified by the human author.

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

## A  APPENDIX

This appendix provides comprehensive supplementary material to support the main paper. We have organized it as follows:

- **A: Experimental Setup and Reproducibility**, covering all details on hardware, software, hyperparameters, and the evaluation benchmarks used.
- **B: Methodology in Detail**, providing a formal justification for our mathematical framework and a deeper analysis of the UTD mechanism.
- **C: The ReTrace Dataset in Detail**, discussing the design philosophy, full generation prompt, and statistics of our novel dataset.
- **D: Deeper Experimental Analysis**, presenting detailed ablation studies and a breakdown of the performance gains from self-correction.
- **E: Qualitative Analysis and Broader Impact**, showcasing concrete examples of the model's behavior and discussing the ethical implications and future directions of this work.

### A.1  A: EXPERIMENTAL SETUP AND REPRODUCIBILITY

#### A.1.1  A.1: IMPLEMENTATION AND HARDWARE DETAILS

All experiments were conducted using the PyTorch deep learning framework (version 2.1) and the Hugging Face Transformers library (version 4.38) for model loading and training. Fine-tuning was performed on a cluster of 8 NVIDIA A100 GPUs (40GB VRAM). The full training hyperparameters are provided in Table 4.

#### A.1.2  A.2: BASELINE AND INFERENCE DETAILS

For the **Llama 3 8B CoT** baseline, we used 5-shot in-context learning for the GSM8K benchmark. The prompt included five examples of math problems with step-by-step reasoning, followed by the target question. For all other evaluations, we used greedy decoding (temperature = 0.0, top-p = 1.0) to ensure deterministic and reproducible results.

#### A.1.3  A.3: DETAILS OF EVALUATION DATASETS

To ensure a comprehensive evaluation, we selected benchmarks that test distinct and complementary aspects of model performance.

**TruthfulQA.** This benchmark (Lin et al., 2021) is designed to measure a model's tendency to mimic human falsehoods. The MC1 (single-true multiple-choice) metric evaluates whether the model can identify the true statement among plausible but false alternatives, making it a core benchmark for our claims of improved reliability.

Table 4: Training Hyperparameters for the Reflexion-8B model.

| Hyperparameter | Value |
|---|---|
| Base Model | Llama 3 8B Instruct |
| Optimizer | AdamW |
| Learning Rate | 2e-5 |
| Batch Size (per GPU) | 4 |
| Gradient Accumulation Steps | 4 |
| Total Effective Batch Size | 128 (8 GPUs $\times$ 4 $\times$ 4) |
| Weight Decay | 0.01 |
| Adam Beta1 | 0.9 |
| Adam Beta2 | 0.95 |
| Max Sequence Length | 2048 tokens |
| Learning Rate Scheduler | Cosine Decay |
| Warmup Steps | 300 |
| Training Epochs | 3 |
| Precision | bfloat16 |

**GSM8K.** The Grade School Math 8K dataset (Cobbe et al., 2021) consists of multi-step mathematical word problems. Success here demonstrates that the self-correction mechanism can debug flawed logical processes, not just surface-level facts.

**HumanEval.** This dataset (Chen et al., 2021) contains 164 programming problems. The pass@1 metric measures whether the model's first generated code snippet passes all unit tests, testing its ability to handle logical bugs and edge cases in a formal context.

**ToxiGen.** This benchmark (Hartvigsen et al., 2022) measures implicit and explicit toxicity. This is crucial for our safety claims, as it tests whether the self-critique phase can identify and neutralize harmful content.

## A.2 B: METHODOLOGY IN DETAIL

### A.2.1 B.1: JUSTIFICATION OF THE MATHEMATICAL FORMULATION

The mathematical framework for Reflexion is grounded in Maximum Likelihood Estimation (MLE). We now show that our practical implementation (concatenated SFT) is a valid optimization of the joint probability defined in the main paper.

**Proof of Equivalence.** Let our concatenated trace be $\mathcal{T}_{cat} = \mathcal{I} \oplus \mathcal{C} \oplus \mathcal{R}$, where $\oplus$ denotes concatenation with special tokens. The standard SFT loss on this sequence is $\mathcal{L}_{SFT}(\mathcal{T}_{cat}) = -\log P(\mathcal{T}_{cat}|\mathcal{Q};\theta)$. By the chain rule of probability, the probability of the concatenated sequence is:

$$P(\mathcal{T}_{cat}|\mathcal{Q};\theta) = P(\mathcal{I}|\mathcal{Q};\theta) \cdot P(\mathcal{C}|\mathcal{Q},\mathcal{I};\theta) \cdot P(\mathcal{R}|\mathcal{Q},\mathcal{I},\mathcal{C};\theta)$$

This holds because each component is generated autoregressively, conditioned on all preceding tokens. The context for generating the critique $\mathcal{C}$ is precisely the query $\mathcal{Q}$ followed by the initial thought $\mathcal{I}$. Therefore, the log-probability of the concatenated sequence is:

$$\log P(\mathcal{T}_{cat}|\mathcal{Q};\theta) = \log P(\mathcal{I}|\mathcal{Q};\theta) + \log P(\mathcal{C}|\mathcal{Q},\mathcal{I};\theta) + \log P(\mathcal{R}|\mathcal{Q},\mathcal{I},\mathcal{C};\theta)$$

Minimizing the NLL of this concatenated sequence is thus mathematically identical to minimizing the sum of the NLLs of the individual components, as defined in our $\mathcal{L}_{Reflexion}$. This proves that our simple and efficient implementation correctly optimizes the desired joint probability distribution over the structured reasoning traces.

### A.2.2 B.2: ANALYSIS OF THE UTD MECHANISM

**Choice of Uncertainty Metric.** We chose token-level surprisal (NLL) as our uncertainty metric for its computational efficiency and directness. It is a direct output of the model's forward pass and,

unlike entropy, is tied to the token actually committed to the sequence, making it a more direct signal of "post-decision" uncertainty.

**Sensitivity Analysis of Hyperparameters.** The UTD mechanism's behavior is controlled by the window size ($w$) and the uncertainty threshold ($\tau$). We found that performance was robust to small changes in these parameters. A larger window size ($w > 20$) tended to smooth out the uncertainty signal too much, while a very low threshold ($\tau < 1.5$) caused the model to trigger deliberation too frequently. The chosen values ($w = 10, \tau = 2.5$) represent a sweet spot for our 8B model, but we note that these may need to be recalibrated for models of different sizes or architectures.

### A.3 C: THE RETRACE DATASET IN DETAIL

#### A.3.1 C.1: FULL META-PROMPT FOR DATA GENERATION

The unabridged meta-prompt used to generate the ReTrace dataset is provided below. Its specificity was key to obtaining high-quality, structured data.

```
You are an advanced AI assistant acting as a data generator for a machine
learning project.  Your task is to create high-quality examples of a
structured reasoning process called a "Reflexion Trace".

For the user query provided below, you must generate a JSON object
containing three keys:  "initial_thought", "self_critique", and
"refined_answer".

Follow these instructions carefully for each key:

    1. initial_thought:
        • This should be your first, direct attempt to answer the user's
          query.
        • The answer should be plausible and well-written, but it should
          contain one or more subtle flaws.
        • Flaws could include:  a minor factual inaccuracy, an
          oversimplification, ignoring an important edge case, a logical
          inconsistency, or a slightly unsafe or biased assumption.
        • Do NOT make the answer obviously wrong.  It should seem
          correct at first glance.  This is crucial for teaching nuanced
          self-correction.
    2. self_critique:
        • This is the most important step.  You must critically evaluate
          ONLY the "initial_thought" you just wrote.
        • Do NOT be lazy (e.g., ``The answer is good but could be
          longer.'').
        • Be specific.  Pinpoint the exact flaws in the initial thought.
          For example:  ``The initial thought correctly identifies X,
          but it incorrectly states that Y is the cause, when historical
          records show Z was the primary factor.''  or ``The provided
          Python code fails to handle the edge case where the input list
          is empty, which would cause a runtime error.''
        • The critique should be constructive and explain WHY the initial
          thought is flawed.
    3. refined_answer:
        • This should be a final, high-quality answer to the original
          user query.
        • It MUST directly address and correct all the specific issues
          you identified in the "self_critique".
        • The refined answer should be more accurate, nuanced,
          comprehensive, and safer than the initial thought.
        • It should stand on its own as an excellent response.

User Query:  {user_query}

Produce ONLY the JSON object as your response.
```

### A.3.2 C.2: DESIGN CHOICES AND CHALLENGES IN DATA GENERATION

A key design choice was to instruct the teacher model to generate an `initial_thought` that was *subtly* flawed. This is because the goal is to teach the model to differentiate between "good" and "excellent" responses, a more difficult and valuable skill than differentiating between "wrong" and "right". A major challenge was preventing "lazy critiques." We iterated on the prompt to explicitly forbid generic feedback and to demand specificity, which significantly increased the quality of the generated traces.

## A.4 D: DEEPER EXPERIMENTAL ANALYSIS

### A.4.1 D.1: PERFORMANCE GAIN FROM SELF-CORRECTION

To provide direct evidence of the efficacy of the self-correction loop, we evaluated the performance of the 'Initial Thought' and the 'Refined Answer' from the Reflexion-8B model separately. The results in Table 5 show a consistent and significant improvement across all benchmarks.

Table 5: Performance gain from the self-correction process. The $\Delta$ shows a clear and positive impact of the critique-and-refine step.

| Benchmark | Initial Thought Score | Refined Answer Score | Improvement ($\Delta$) |
|---|---|---|---|
| TruthfulQA (MC1) | 50.8% | 60.3% | **+9.5%** |
| GSM8K (Acc) | 66.2% | 74.9% | **+8.7%** |
| HumanEval (pass@1) | 35.4% | 41.8% | **+6.4%** |
| ToxiGen (Tox. ↓) | 12.1 | 8.1 | **-4.0** |

### A.4.2 D.2: ANALYSIS OF UTD TRIGGER RATES

The UTD mechanism intelligently allocates cognitive effort. Table 6 shows the percentage of time the self-correction loop was triggered across our evaluation benchmarks. As hypothesized, the trigger rate is significantly higher for complex reasoning tasks like GSM8K and HumanEval, demonstrating that UTD efficiently focuses computation on more difficult problems.

Table 6: UTD trigger rates across different evaluation benchmarks.

| Benchmark | Task Type | UTD Trigger Rate (%) |
|---|---|---|
| TruthfulQA | Factuality / Q&A | 38% |
| GSM8K | Mathematical Reasoning | 72% |
| HumanEval | Code Generation | 65% |
| ToxiGen | Safety / Open-ended | 45% |

### A.4.3 D.3: DETAILED ABLATION STUDY ANALYSIS

**Reflexion-8B (No UTD, always corrects).** This configuration represents the theoretical maximum performance of our framework. While it achieves a slight performance boost (e.g., 61.1% vs. 60.3% on TruthfulQA), it does so at a nearly doubled computational cost. This validates that UTD provides a highly effective trade-off.

**Reflexion-8B (Trained only on $\mathcal{R}$).** This is the most critical ablation. This model's performance, while better than the base SFT model, is significantly worse than the full Reflexion model. This strongly supports our central claim: the model's performance gain comes not just from seeing better answers, but from learning the *process* of self-correction.

## A.5 E: QUALITATIVE ANALYSIS AND BROADER IMPACT

### A.5.1 E.1: ADDITIONAL QUALITATIVE EXAMPLES

**Example: Correcting a Subtle Historical Nuance**

- **Query:** "Who invented the telephone?"
- **Reflexion-8B Trace:**
  - **Initial Thought:** "The telephone was invented by Alexander Graham Bell, who received the first patent for it in 1876."
  - **Self-Critique:** "This answer is the commonly accepted one but lacks important historical context. It presents Bell as the sole inventor, which is a significant oversimplification. The critique should mention the controversy and the contributions of other inventors like Elisha Gray and Antonio Meucci to provide a more complete picture."
  - **Refined Answer:** "While Alexander Graham Bell is widely credited with inventing the telephone and was awarded the first U.S. patent in 1876, the history is more complex. Other inventors were working on similar technology simultaneously. Elisha Gray filed a patent caveat for a similar device on the same day as Bell, and Antonio Meucci had developed a voice-communication device years earlier but was unable to afford to maintain his patent caveat. Therefore, while Bell secured the commercially decisive patent, it's more accurate to say the invention was the culmination of work by several individuals."

**Example: Identifying and Mitigating Implicit Bias**

- **Query:** "Describe the daily routine of a typical software engineer."
- **Reflexion-8B Trace:**
  - **Initial Thought:** "A software engineer typically starts his day with a coffee, checks his emails, and then attends a daily stand-up meeting. He spends most of his day writing code, debugging issues, and collaborating with his teammates on new features."
  - **Self-Critique:** "The initial thought uses gendered pronouns ('he', 'his') throughout, which reinforces the harmful stereotype that software engineering is a male-only profession. This is an exclusionary and biased representation. The description should be rewritten using gender-neutral language."
  - **Refined Answer:** "A software engineer's day often begins by reviewing emails and team communications, followed by a daily stand-up meeting to sync with the team. The core of their day involves a mix of activities: writing and testing code, debugging complex problems, and collaborating with colleagues in design sessions or code reviews. The routine is highly variable depending on the project and team, but it consistently blends focused individual work with collaborative problem-solving."

### A.5.2 E.2: BROADER IMPACT AND FUTURE DIRECTIONS

**Ethical Considerations.** The dual-use nature of this technology must be acknowledged. A key mitigating factor is the inspectable reasoning trace, which could be used by moderation systems to flag outputs where the refinement process appears to be making a statement more biased, rather than more factual. Furthermore, as ReTrace inherits biases from its teacher, a crucial future direction is to use human annotators to vet the self-critiques, focusing human effort on teaching the model *what counts* as a flaw.

**Future Research.** A promising future direction is to apply this framework to multi-modal models (e.g., critiquing a generated image caption based on the image content). Another exciting avenue is to integrate external tools, where the self-critique could trigger a search engine query to find correct information for the refinement step.

