# OpenReview forum: "Reflexion: Language Models that Think Twice for Internalized Self-Correction"
_ICLR.cc/2026/Conference — ICLR 2026 Conference Desk Rejected Submission_

### Official Review · Reviewer_JPEV · 2025-10-30

**Soundness:** 2
**Presentation:** 2
**Contribution:** 2
**Rating:** 2
**Confidence:** 4

**Summary:**

This paper proposes Reflexion, a framework that teaches language models to self-correct by explicitly learning a three-step reasoning process: generate, critique, and refine.
Using a new dataset called ReTrace, which contains 200k structured examples of self-correction generated by GPT-4o, the model learns to improve its own reasoning internally.
Experiments show that a Reflexion-trained 8B model outperforms larger baselines on factuality and reasoning benchmarks, indicating that supervising the reasoning process itself can be more effective than simply scaling model size.

**Strengths:**

The paper presents a novel framework, Reflexion, that enables language models to perform explicit self-correction through a structured “generate–critique–refine” reasoning process.
It introduces ReTrace, a large-scale dataset of 200k self-correction traces, allowing the model to internalize reflective reasoning rather than rely solely on outcome supervision.
Empirical results show that an 8B Reflexion model achieves significant gains in factuality and reasoning, even outperforming a 70B baseline, demonstrating impressive parameter efficiency and reliability.

**Weaknesses:**

- The paper does not conduct a rigorous comparison with other methods that explicitly learn reasoning processes, such as CoT, which is based on approaches. Therefore, it remains unclear whether the proposed method is actually superior to these existing techniques.
- Because the proposed method includes an additional refinement stage, its training cost is considerably higher. The same computational budget could be used to train baseline models with additional data instead. Since the paper does not provide results under equal computational conditions, its practical efficiency is uncertain.

**Questions:**

- It is unclear why training only on the final answers leads to such a large degradation in performance. This setting corresponds to standard imitation learning, and such a substantial drop appears somewhat unnatural.

---

### Official Review · Reviewer_aapg · 2025-10-31

**Soundness:** 2
**Presentation:** 1
**Contribution:** 1
**Rating:** 2
**Confidence:** 5

**Summary:**

This paper introduces Reflexion, a framework to teach language models an internalized skill of self-correction. The core idea is to train a model on a three-step reasoning trace: generate -> critique -> refine, distilled from a teacher model. The authors make three primary contributions: (1) A simple SFT method on teacher traces, (2) a new dataset, ReTrace, distilled from GPT-4o, and (3) an efficient inference mechanism, Uncertainty-Triggered Deliberation (UTD).

**Strengths:**

- The paper is generally easy to follow.
- The SFT approach is simple and seems effective.
- The ReTrace dataset could be a useful contribution for research on process supervision.

**Weaknesses:**

The paper's potential contributions are undermined by multiple, critical flaws in its literature review, evaluation, and claims.
- The paper's primary flaw is its complete failure to engage with the relevant literature from the last two years (the related work section stop in 2023). LLM Self-correction is an active research area, and this omission leads to unsubstantiated claims of novelty. For example, the paper frames Reflexion as a "new paradigm," but this claim is unsupportable. It ignores past methods (e.g., RISE, NeurIPS 2024) that already explore iterative refinement, and it fails to position its "single-shot" trace against these more general approaches. The paper advocates for SFT and critiques RL. However, it fails to cite or compare against recent work (e.g., SCoRe, ICLR 2025) that directly argues offline SFT (like Reflexion's) is "insufficient" and that an online RL approach is necessary.
- The empirical results are not strong enough to support the paper's claims. The qualitative analysis is anecdotal, not analytical. The corresponding section consists of only four lines and a single example, which provides no insight into the model's failure modes or the general quality of its self-correction. The performance gains on UTD are marginal.
- The presentation of results is unorthodox, for example, the tables bold results that are not the best.

ref
- [SCoRe, ICLR 2025] Training Language Models to Self-Correct via Reinforcement Learning
- [RISE, NeurIPS 2024] Recursive Introspection: Teaching Language Model Agents How to Self-Improve

**Questions:**

I suggest the authors conduct a new, thorough literature search for 2024-2025 papers on self-correction, self-refinement, and process supervision. Consider rewrite the Introduction and Related Work sections to accurately position the paper against the actual state-of-the-art and re-frame the paper's central claim.

---

### Official Review · Reviewer_s46B · 2025-10-31

**Soundness:** 4
**Presentation:** 3
**Contribution:** 3
**Rating:** 4
**Confidence:** 4

**Summary:**

The paper introduces Reflexion, a framework that trains language models to self-correct through a three-step process—generate, critique, and refine. Using a large synthetic dataset and an uncertainty-based trigger, it improves factuality and reasoning.

**Strengths:**

1. The paper is clearly written and well organized.

2. The proposed method is conceptually straightforward and easy to implement.

**Weaknesses:**

1. From my perspective, the approach mainly combines ideas from knowledge distillation and prior works such as [1,2], which limits the contribution of this paper.

2. The ReTrace dataset is part of the contribution. However, the human verification of this generated dataset is quite limited, raising some concerns about its overall quality and reliability.

3. The paper would be stronger with comparisons to closely related methods like ToT, STaR, or Self-Refine, which share similar motivations.

4. The analysis of hyperparameters is rather brief; adding experimental evidence or ablation results would help substantiate the discussion.

```
[1] Yao, Shunyu, et al. "Tree of thoughts: Deliberate problem solving with large language models." Advances in neural information processing systems 36 (2023): 11809-11822.
[2] Lightman, Hunter, et al. "Let's verify step by step." The Twelfth International Conference on Learning Representations. 2023.
```

**Questions:**

Please see the Weaknesses part.

---

### Official Review · Reviewer_etWC · 2025-11-01

**Soundness:** 2
**Presentation:** 2
**Contribution:** 2
**Rating:** 4
**Confidence:** 4

**Summary:**

This paper introduces reflexion, a framework aimed at reducing hallucinations in large language models (llms) by instilling an internalized self-correction capability. The authors contribute a large-scale dataset of 200,000 structured self-correction traces, bootstrapped from gpt-4o, and an inference-time mechanism that activates the full critique-refine loop selectively based on the model's detected uncertainty in its initial thought.

**Strengths:**

The paper demonstrates that an 8b parameter model fine-tuned with reflexion significantly outperforms its baseline counterparts and even rivals the performance of a much larger 70b model on key benchmarks like truthfulqa and gsm8k, highlighting impressive parameter efficiency.

**Weaknesses:**

1.  The paper's review of related work is limited, primarily citing pre-2023 literature. it fails to adequately situate itself among numerous existing works exploring similar generate-critique-refine paradigms (e.g., self-refine, crt, self-rag), thereby undermining its claim of being a "new paradigm."

2.
- The primary result presented in Table 2 combines the effects of fine-tuning on ReTrace with the use of UTD at inference. It is impossible to determine how much of the performance gain originates from the dataset (ReTrace) versus the inference-time mechanism (UTD), respectivly.
- In Table 3,the model trained solely on the final refined answers (`Trained only on R`) yields a score of 52.5% on TruthfulQA, which is very close to the Llama 3 8B CoT baseline score of 51.3%. Does this indicate that the primary improvement in performance does not stem from the ReTrace dataset providing superior answer knowledge, but rather from the UTD-activated self-correction process during inference? This calls into question the dataset's claimed contribution.
- If the ReTrace dataset is a significant contribution, its effectiveness should be demonstrated by fine-tuning models from various  models (e.g., Gemma, Mistral), not just Llama 3. Its generalizability remains unproven.
- The efectiveness of  the framework (Table2) lacks comparisons with other models as  baselines. It remains untested whether UTD can be applied as a plug-in inference strategy for other models, which is essential for establishing its general utility.

**Questions:**

Please refer to the above weaknesses.

---

### Official Review · Reviewer_ksNr · 2025-11-01

**Soundness:** 2
**Presentation:** 2
**Contribution:** 3
**Rating:** 4
**Confidence:** 3

**Summary:**

This paper proposes Reflexion, a process-supervised framework that teaches an LLM to “think twice” by explicitly generating a three-part trace (Initial Thought , Self-Critique, Refined Answer), using SFT on concatenated sequences. It also introduces ReTrace, a 200k-example dataset of structured self-correction traces bootstrapped from a teacher model, and an inference policy Uncertainty-Triggered Deliberation (UTD) that engages the critique-and-refine loop only when token-level surprisal signals high uncertainty. On multiple datasets spanning factuality, reasoning, and safety, an 8B model trained with Reflexion improves substantially over 8B baselines and approaches a 70B model, while UTD balances quality and compute.

**Strengths:**

* The UTD approach, which dynamically triggers self-correction based on average uncertainty over recent tokens, is quite interesting.
* The method shows good parameter-efficiency and strong results in improving accuracy, with 8B model trained on the dataset closing the gap on factuality and reasoning against a 70B baseline.
* The ablations (R-only, no UTD and always correct) and the sensitivity analysis on window size and the uncertainty threshold are very helpful (although specific results of the sweeps are not reported).

**Weaknesses:**

* The paper claims the approach is "a more data-efficient path to reliable reasoning". However, it does not provide clear evidence or analysis on data-efficiency. Also, it relies on a strong teacher LLM to generate high-quality critiques, which can be costly and unscalable in practice.
* Critically, the comparisons are against standard SFT 8/70B, or with CoT. There are no head-to-head evaluation against any other process-supervised or self-deliberation /self-refine style training method (a large body of works, e.g. [1-7]).
* (Related to the last point) The paper claims that "ReTrace stands as the first large-scale dataset specifically designed to teach the process of internalized self-correction". I believe this over-claims the contribution. There are well-established, similar datasets that incorporate feedback/critic and refinement, both in general domains (e.g. CriticBench [8]) or specific domains like Math reasoning  (e.g. [4, 9]) or safety (e.g. [10]).
* The most informative ablation (R-only) is helpful, but missing controls include: (1). training on I+R without C (to test whether critique text is essential versus just longer supervision), and (2). a two-model variant (separate critic/solver models instead of the same backbone) to disentangle performance gains from joint vs. modular training.
---

[1]. Learning to Refine with Fine-Grained Natural Language Feedback

[2]. STaR: Bootstrapping Reasoning With Reasoning

[3]. Self-Refine Instruction-Tuning for Aligning Reasoning in Language Models

[4]. Let's Verify Step by Step

[5]. Process Reward Models That Think

[6]. Self-Generated Critiques Boost Reward Modeling for Language Models

[7]. Recursive Introspection: Teaching Language Model Agents How to Self-Improve

[8]. CRITICBENCH: Benchmarking LLMs for Critique-Correct Reasoning

[9]. ProcessBench: Identifying Process Errors in Mathematical Reasoning

[10]. Training Socially Aligned Language Models on Simulated Social Interactions

**Questions:**

* Since UTD hinges on uncertainty, can you report uncertainty-related metrics such as ECE / Brier score and AURC? This would show whether Reflexion improves not just accuracy but reliability under selective prediction.
* What are the specific results for sweeping w and tau?
* Nitpick: I would avoid (intentionally) using the same name from a well-established prior work ("Reflexion"), although this can be fairly minor and depend on personal taste.

---

### Note · Program_Chairs · 2026-01-17
**Submission Desk Rejected by Program Chairs**

The following references in this submission do not refer to real documents and/or have major errors in bibliographic information:

 Guang-He Xiao, Haolin Wang, and Yong-Feng Zhang. Rethinking uncertainty in llms: A case study on a fact-checking benchmark. arXiv preprint arXiv:2305.11382, 2023